# TEXT-DRIVEN EDITING OF 3D SCENES WITHOUT RETRAINING

## ABSTRACT

Numerous diffusion models have recently been applied to image synthesis and editing. However, editing 3D scenes is still in its early stages. It poses various challenges, such as the requirement to design specific methods for different editing types, retraining new models for various 3D scenes, and the absence of convenient human interaction during editing. To tackle these issues, we introduce a text-driven editing method, termed DN2N, which allows for the *direct* acquisition of a NeRF model with universal editing capabilities, eliminating the requirement for retraining. Our method employs off-the-shelf text-based editing models of 2D images to modify the 3D scene images, followed by a filtering process to discard poorly edited images that disrupt 3D consistency. We then consider the remaining inconsistency as a problem of removing noise perturbation, which can be solved by generating data with similar perturbation characteristics for training. We propose cross-view regularization terms to help the DN2N model mitigate these perturbations. Our text-driven method allows users to edit a 3D scene with their desired description, which is more friendly, intuitive, and practical than prior works. Empirical results show that our method achieves multiple editing types, including but not limited to appearance editing, weather transition, object changing, and style transfer. Most significantly, our method exhibits strong generalization of editing capabilities, eliminating the need to customize or retrain editing models for specific scenes or editing types. While reducing editing time and memory overhead, our approach realizes visual outcomes on par with or exceeding previous techniques needing iterative optimization.

## 1 INTRODUCTION

Significant advances in neural radiance field (NeRF) techniques (Mildenhall et al., 2020; Yu et al., 2021; Müller et al., 2022; Barron et al., 2021; Zhang et al., 2020; Wang et al., 2021; Lin et al., 2021; Chen et al., 2022) have been modeled for a variety of vision tasks including novel view synthesis and editing. Numerous NeRF-based methods have been developed to achieve specific types of 3D manipulating, such as appearance editing (Wang et al., 2022a; Martin-Brualla et al., 2021; Kobayashi et al., 2022), scene composition (Tang et al., 2022; Tancik et al., 2022), weather transformation (Li et al., 2022b), multiple editing (Fang et al., 2022; 2023), and style transfer (Huang et al., 2022; Gu et al., 2022; Nguyen-Phuoc et al., 2022; Fan et al., 2022). Recently, a few attempts have been made to leverage multi-modal techniques to design text-guided 3D editing methods (Zhang et al., 2022; Haque et al., 2023). Despite the demonstrated success, several challenges remain. 1) existing methods typically rely on known editing types in advance, resulting in limited modification capabilities; 2) retraining an editing model is required for each particular 3D scene, leading to computational and memory overhead; 3) these techniques are often less user-friendly.

In this work, we propose a novel 3D scene editing method to tackle the challenges above. We consider designing a text-driven and generalized method that involves editing the images of a 3D scene using the off-the-shelf 2D image editing models (Saharia et al., 2022; Rombach et al., 2022; Hertz et al., 2022; Brooks et al., 2022; Mokady et al., 2022), followed by reconstructing the edited scene using a generalizable NeRF model. Nonetheless, creating such an editing pipeline presents several challenges. For example, applying 2D-image editing directly makes it challenging to achieve 3D consistency, and the degree of 3D inconsistency may vary depending on the scene and editing

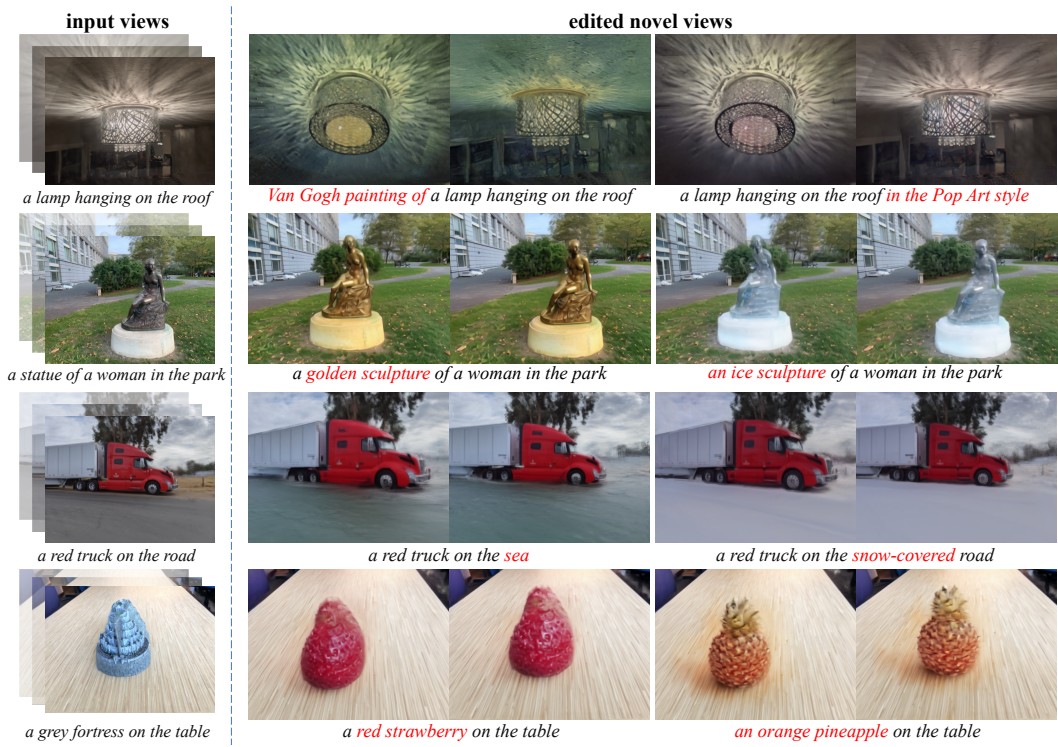

|  input views  |  edited novel views  |

*a lamp hanging on the roof*   —   *Van Gogh painting of* a lamp hanging on the roof   —   a lamp hanging on the roof *in the Pop Art style*

*a statue of a woman in the park*   —   a *golden sculpture* of a woman in the park   —   *an ice sculpture* of a woman in the park

*a red truck on the road*   —   a red truck on the *sea*   —   a red truck on the *snow-covered* road

*a grey fortress on the table*   —   a *red strawberry* on the table   —   *an orange pineapple* on the table

Figure 1: **Visualization of editing results**. The proposed DN2N method enables users to obtain realistic and 3D consistent novel views that match the text caption without retraining a new model.

type. Furthermore, creating a generalized model that does not require retraining for different scenes and editing types necessitates abundant task-specific 3D scene data (see Figure 1).

To address the aforementioned issues, we initially utilize a 2D editing model to perform the preliminary editing on the images of a 3D scene. We subsequently apply a designed content filter to remove images with poor editing results that cause significant 3D inconsistency. However, the remaining images may still contain inconsistent 3D results, in which we consider them as noisy perturbations to the consistently edited images due to the inherent stochastic and diverse nature of the 2D editing model. To tackle this issue, we leverage the perturbation characteristic to create training data pairs by generating image captions through the BLIP model (Li et al., 2022a; 2023) and target captions via GPT (Brown et al., 2020a), then applying minor perturbations associated with these captions to a 3D scene. Therefore, these perturbations can be viewed as noise, as well as unedited images as pseudo ground truth. We further introduce two cross-view regularization terms during training, including the self and neighboring views, to improve the 3D editing consistency. The former requires the NeRF model to generate consistent results for the same target view that is derived from two different source views, while the latter enforces the overlapping pixel values between the target and adjacent views to be approximately close. Finally, both the perturbation dataset and regularization terms are incorporated into our generalizable NeRF model training to facilitate its 3D consistency.

The main contributions of this work are:

- We develop a versatile text-driven 3D scene editing framework named DN2N that employs off-the-shelf 2D editing models for 3D scene manipulation, where the induced 3D inconsistency is modeled as noise perturbation and addressed by generating training data with similar perturbation characteristics for optimization.

- We design a generalizable NeRF model architecture and integrate cross-view regularization terms into the training process to enhance the 3D consistency in edited novel views.

- We conduct extensive experiments of different editing types on multiple datasets. Compared with other approaches, DN2N offers diverse editing capabilities within a shorter

time and lower memory overhead, as well as eliminating the necessity for customizing or retraining a model for different scenes and editing types.

## 2 RELATED WORK

**NeRF-based novel view synthesis**. Novel view synthesis based on NeRF has recently gained significant attention in the vision and graphics communities (Mildenhall et al., 2020; Fridovich-Keil et al., 2022; Müller et al., 2022; Chen et al., 2022; Lin et al., 2021; Fang et al., 2023). NeRF represents the structure and appearance of a 3D scene by using a neural network that takes the spatial location and view direction as input and outputs the corresponding color and opacity at each pixel. Subsequent works have improved NeRF, such as speeding up the training process (Müller et al., 2022; Chen et al., 2022), designing better sampling strategies (Barron et al., 2021), and enhancing generalization ability (Wang et al., 2021; Johari et al., 2022; Liu et al., 2022; Chen et al., 2021).

**Diffusion-based image editing**. Diffusion-based models have been widely used in image generation (Sohl-Dickstein et al., 2015; Dhariwal & Nichol, 2021; Ho et al., 2020; Saharia et al., 2022; Ramesh et al., 2022; Rombach et al., 2022; Song et al., 2021a;b). Numerous text-based image editing methods have recently been developed, such as GLIDE (Nichol et al., 2022) and Stable Diffusion (Rombach et al., 2022). Imagic (Kawar et al., 2022) finetunes images according to text descriptions, and Prompt-to-prompt (Hertz et al., 2022) preserves unedited regions by utilizing cross-attention information. Pix2pix-zero (Parmar et al., 2023) employs embedding vector mechanisms to establish controllable editing directions for images. InstructPix2Pix (Brooks et al., 2022) trains a model by generating a large number of text-editing image pairs using GPT (Brown et al., 2020b) and Stable Diffusion. Null-text inversion (Mokady et al., 2022) proposes a more accurate diffusion inversion process to enhance image-controlled editing capabilities.

**3D scene editing**. Numerous 3D scene editing approaches have been developed based on point cloud representations (Huang et al., 2021; Mu et al., 2022a), and scene texture mapping based on triangle meshes (Zhou & Koltun, 2014; Höllein et al., 2022; Han et al., 2021). However, these methods are limited by the inherent scene representations, which restrict scalability and editing capability. Recently, NeRF-based methods have been used for 3D scene editing, including appearance editing (Boss et al., 2021a;b; Li et al., 2022c; Rudnev et al., 2021; Kobayashi et al., 2022; Tschernezki et al., 2022), scene combining (Fridovich-Keil et al., 2022; Tang et al., 2022), weather simulation (Li et al., 2022b), assemble editing (Fang et al., 2022; 2023), and style transfer (Chiang et al., 2022; Huang et al., 2022; Gu et al., 2022; Mu et al., 2022b; Wang et al., 2022b; Zhang et al., 2022). However, several aspects of existing 3D editing methods based on NeRF are limited. For instance, they are usually restricted to performing a single editing task, necessitating separate model structures and training approaches for each editing capability. To overcome this limitation, PVD-AL (Fang et al., 2023) employs distillation to achieve multiple editing capabilities in one model, although with lower efficiency. Instruct-N2N (Haque et al., 2023) enables controllable editing of 3D scenes by pre-editing 2D images using InstructPix2Pix (Brooks et al., 2022). However, Instruct-N2N has two notable drawbacks. First, retraining is necessary for each new editing direction, resulting in significant computation and memory overhead. Second, the method generates new training data and updates model parameters during training, which is time-consuming and has high memory overhead. In contrast, our method overcomes these two problems by requiring only the target text to obtain the corresponding 3D scene editing, using the same model for any editing types without retraining, thereby reducing model storage consumption and training time.

## 3 PROPOSED METHOD

Figure 2 shows the overall framework of our method. At the training stage, we first utilize BLIP (Li et al., 2023) and GPT (Brown et al., 2020a) models to generate the input and target captions of a scene. We then use a 2D image editing model (Mokady et al., 2022) to apply minor editing perturbations to the scene based on these captions to obtain training data pairs, while the training objective of our model is to remove these perturbations (see Eq. 4). We use two sets of independent source views to render the same target view, resulting in $Tgt_a$ and $Tgt_b$, respectively, and generate a neighboring view around the target view to obtain $Nbr$. Then, we impose a consistency loss (see Eq. 5 and Eq. 6) on the three rendered results. At the inference stage, we begin by applying standard

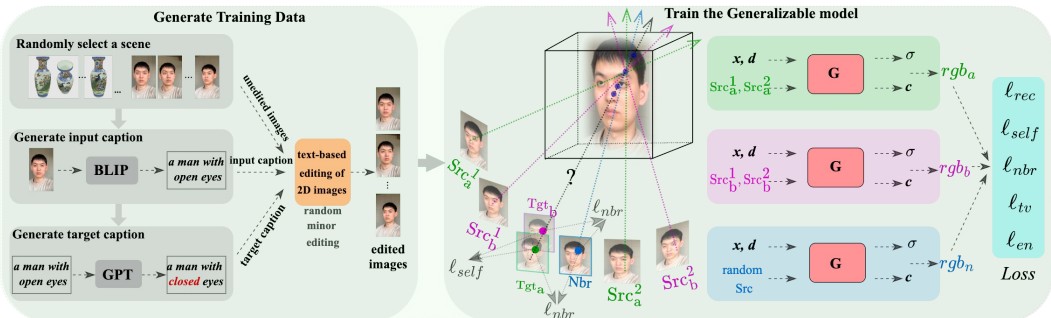

(a) **Training stage**. We generate training data by applying subtle perturbations to 3D scenes using the BLIP (Li et al., 2023), GPT (Brown et al., 2020a), and 2D editing models (Mokady et al., 2022). Subsequently, we train the generalized NeRF model $G$ by incorporating cross-view regularization terms, $\ell_{self}$ and $\ell_{nbr}$.

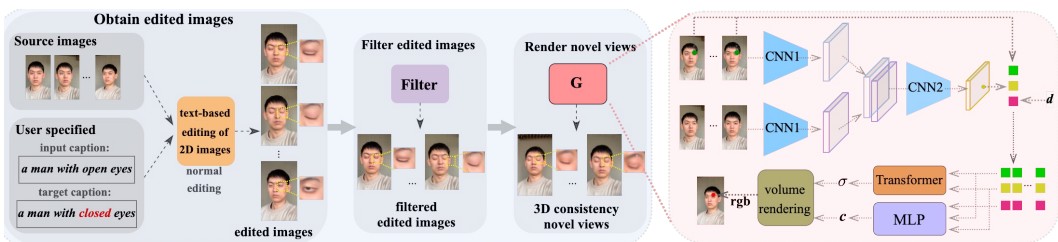

(b) **Inference stage**. We devise a content filter to eliminate the images with subpar editing results and compromised consistency. Then, we utilize the well-trained model to obtain the edited novel views.

Figure 2: **Illustration of the DN2N framework**. See Sec 3 for the detailed description.

magnitude editing to the 3D scene. Subsequently, we filter out images with poor editing effects and those inducing significant disruptions to the 3D consistency (see Eq. 9). Finally, we feed the filtered images into the generalizable NeRF model to obtain edited novel views directly.

**Optimization objectives**. A 3D scene training data consists of $N$ images and their corresponding camera parameters: $\{I_i, P_i\}_{i=1}^N$. To obtain the pre-edited image $\tilde{I}_i$ for each $I_i$, we employ the 2D image editing model, which is defined as:

$$\tilde{I}_i = \mathcal{F}(I_i, C_{in}^i, C_{tgt}^i, \theta), \tag{1}$$

where $C_{in}^i$ is input caption for the unedited image $I_i$, $C_{tgt}^i$ is target caption for the edited image $\tilde{I}_i$, and $\theta$ is the hyper-parameters of model $\mathcal{F}$. After filtering out the poorly edited images, the resulting data is denoted as $\{\tilde{I}_m, P_m\}_{m=1}^M$, where $M < N$. Then the generalized NeRF model $G$ with parameter $\Theta$ predicts the target view $\hat{I}_m$ using $K$ source views $\{\tilde{I}_k, P_k\}_{k=1}^K$:

$$\hat{I}_m = G(\tilde{I}_k, P_k, \Theta \mid k = 1, \ldots, K, P_k \neq P_m). \tag{2}$$

Assuming that the ground truth of edited images with 3D consistency is denoted as $\breve{I}_m$, our optimization objective can be expressed as:

$$\arg\min_{\Theta} \sum_{m=1}^M \left\| \breve{I}_m - \hat{I}_m \right\|^2 = \arg\min_{\Theta} \sum_{m=1}^M \left\| \breve{I}_m - G(\tilde{I}_k, P_k, \Theta \mid k = 1, \ldots, K, P_k \neq P_m) \right\|^2. \tag{3}$$

However, the edited ground truth $\breve{I}_m$ is not at our disposal. Thus, we express the result of Eq. 1 as $\tilde{I}_m = \breve{I}_m + \triangle I_m$. This implies that the images edited by model $\mathcal{F}$ can be perceived as edited ground truth images with minor perturbations caused by noise. In this work, the perturbations mainly stem from the lack of consistency in the training dataset after 2D image editing by model $\mathcal{F}$. Hence, we can apply similar random minor perturbations $\triangle I_i$ to the clean images $\{I_i\}_{i=1}^N$ by controlling the parameters $\theta$ in $\mathcal{F}$, resulting in $\{\tilde{I}_i\}_{i=1}^N$. We use the $\{\tilde{I}_i\}_{i=1}^N$ as pseudo ground truth to train the model $G$. As such, the objective function constructed for model training is:

$$\arg\min_{\Theta} \sum_{i=1}^{N} \left\| I_i - G(\tilde{I}_k, P_k, \Theta \mid k = 1, \ldots, K, P_k \neq P_i) \right\|^2. \tag{4}$$

**Generate training data**. To generate training data pairs as shown in Figure 2, we first select an image from a scene at random and feed it into the BLIP (Li et al., 2023) model to obtain its input caption, such as "a man with open eyes." Subsequently, we use GPT (Brown et al., 2020a) to generate the target caption, such as "a man with closed eyes." For each image of a scene, we apply minor perturbations randomly using Eq. 1 by controlling the hyper-parameters $\theta$. More details of the generation process are available in the supplementary material.

**Self-view robustness.** We use a training approach similar to that in the generalizable NeRF models (Wang et al., 2021; Liu et al., 2022; Johari et al., 2022), which involves predicting the target view based on several source views. When the training data of a 3D scene is consistent, predicting the same target view with different source views typically yields consistent results. However, this may not hold for the scene pre-edited by a 2D-editing model. To address this, we perform two independent predictions, labeled $\mathcal{A}$ and $\mathcal{B}$. For $\mathcal{A}$, we use source views $\{Src_a^1, Src_a^2, ...\}$ and predict the target view as $Tgt_a$. In addition, for $\mathcal{B}$, we use source views $\{Src_b^1, Src_b^2, ...\}$ and predict the same target view as $Tgt_b$. We calculate the L2 loss to ensure consistency between two predictions:

$$\ell_{self} = \|Tgt_a - Tgt_b\|^2. \tag{5}$$

**Neighboring view consistency**. Empirically, there are often noticeable texture or color discontinuities between adjacent views when rendering along a smooth camera path. To tackle this issue, we enforce a smooth transition between adjacent views. Specifically, we slightly perturb the pose corresponding to the target view and generate a neighboring view $Tgt_{nbr}$ based on the perturbed pose. We project the pixel points from the target view onto the neighboring view using depth and calculate the following loss to minimize the image discontinuities caused by changes in the viewing angle:

$$\ell_{nbr} = \|M(Tgt_a - Tgt_{nbr})\|^2 + \|M(Tgt_b - Tgt_{nbr})\|^2, \tag{6}$$

where $M$ refers to a mask, which indicates that only the loss between overlapping regions is calculated. In addition, we note that the weights $w_i$ of some sampled points on the rays are distributed along the rays. This can lead to distortions such as floating objects in the predicted target view, which may affect the accuracy of the target view to neighboring view projection. Similar to (Kim et al., 2022), we introduce an entropy loss for the weights of the sampling points:

$$\ell_{en} = -\sum w_i \log w_i = -\sum T_i(1 - \exp(-\sigma_i \delta_i)) \log (T_i(1 - \exp(-\sigma_i \delta_i))), \tag{7}$$

where $\sigma_i$ is the density of the spatial points sampled along a ray. $\delta_i$ denotes the distance between adjacent sample points. Please refer to the supplementary material for the definition of $T_i$.

**Network structure**. The generalized NeRF model $G$ in Figure 2 is developed based on the IBR-Net (Wang et al., 2021). We extend it by incorporating multi-viewpoint aggregation, cross-view mappings, and integration of unedited image information to render consistent results. More details regarding the generalized NeRF model $G$ are described in the supplementary material.

**Loss function**. During training, the model cannot render a complete image in a single forward pass due to GPU memory limitations. Thus, the model predicts image patches at each training step, and the loss is computed on a patch level. The total loss function employed in this work is:

$$\ell = \ell_{rec} + \lambda_1 \ell_{self} + \lambda_2 \ell_{nbr} + \lambda_3 \ell_{en} + \lambda_4 \ell_{tv}, \tag{8}$$

where $\ell_{rec} = \left\| I_i - \hat{I}_i \right\|^2$, and $\ell_{tv}$ is the total variation regularization term (Rudin & Osher, 1994).

**Content filter**. During the inference phase, we design a content filter to remove poorly edited images and those that cause significant inconsistency. The remaining perturbations caused by 2D editing are then removed using the trained model. Our content filter is designed based on four tuples:

$$\{\text{SSIM}(I_i, \tilde{I}_i), \ \text{CLIP}(I_i, \tilde{I}_i), \ \text{CLIP}(\tilde{I}_i, C_{tgt}), \ \text{CLIP}(I_i, \tilde{I}_i) - \text{CLIP}(C_{in}, C_{tgt})\}_{i=1}^{N}. \tag{9}$$

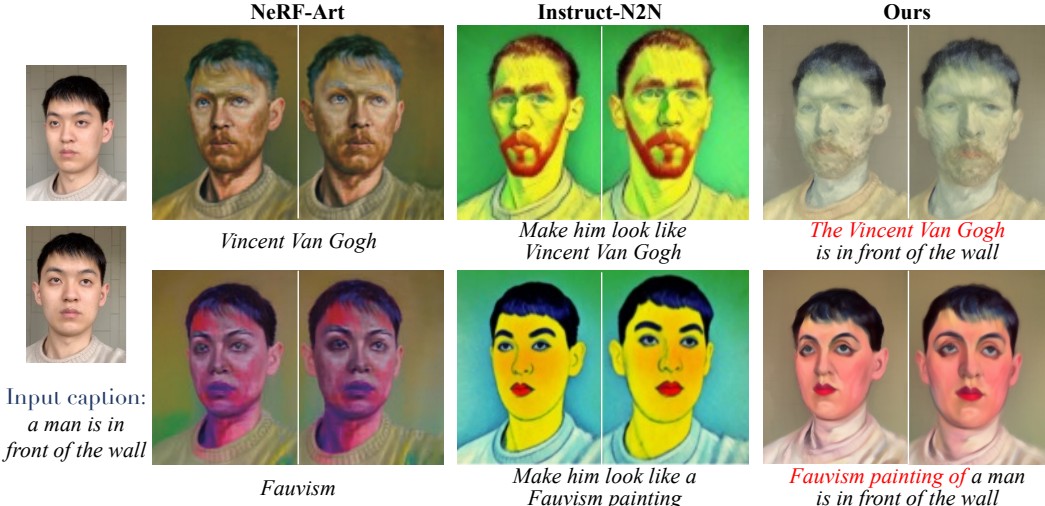

Figure 3: **Comparison with other text-driven editing methods**. DN2N strikes a better balance between preserving image content and aligning with textual descriptions. More importantly, it is not necessary to retrain our model for different editing types.

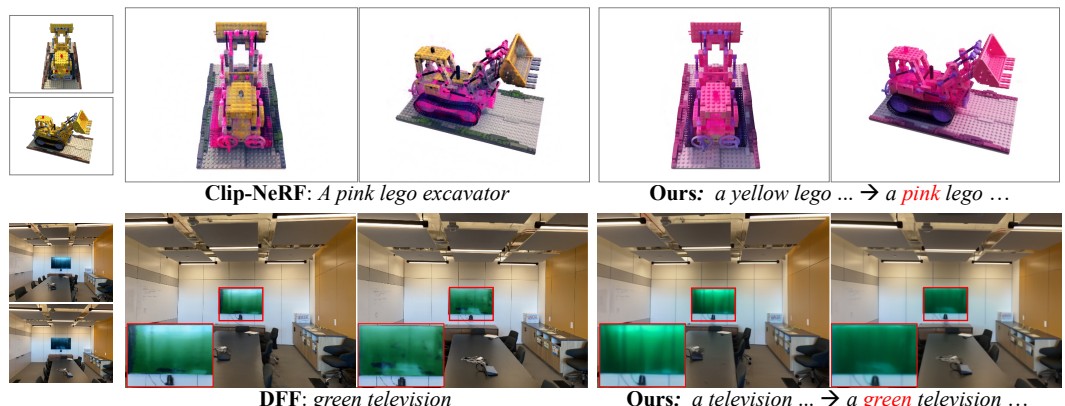

Figure 4: **Comparison with other methods for editing appearance**. DN2N achieves higher accuracy in matching target captions while effectively preserving information in non-edited areas.

We calculate these four groups of values for all the edited images, sort each set of results, and discard the top 10% or bottom 10% of the data. This content filter significantly reduces the 3D inconsistency among the edited images. More implementation details are presented in the supplementary material.

## 4 EXPERIMENTS AND ANALYSIS

The datasets used to train our model are Google Scanned Objects (Downs et al., 2022), NeRF-Synthetic (Mildenhall et al., 2020), Spaces (Flynn et al., 2019), IBRNet-collect (Wang et al., 2021), LLFF (Mildenhall et al., 2019) and NeRF-Art (Wang et al., 2022b). The default 2D-image editing model is Null-text (Mokady et al., 2022). We implement our method with PyTorch (Paszke et al., 2019), train the model on 8 NVIDIA V100 GPUs, and use one single V100 GPU for inference. More implementation details and results are presented in the supplementary material.

### 4.1 QUALITATIVE RESULTS

As illustrated in Figure 1, our approach demonstrates its ability to achieve various challenging editing types without retraining while maintaining 3D consistency and conforming to the text descrip-

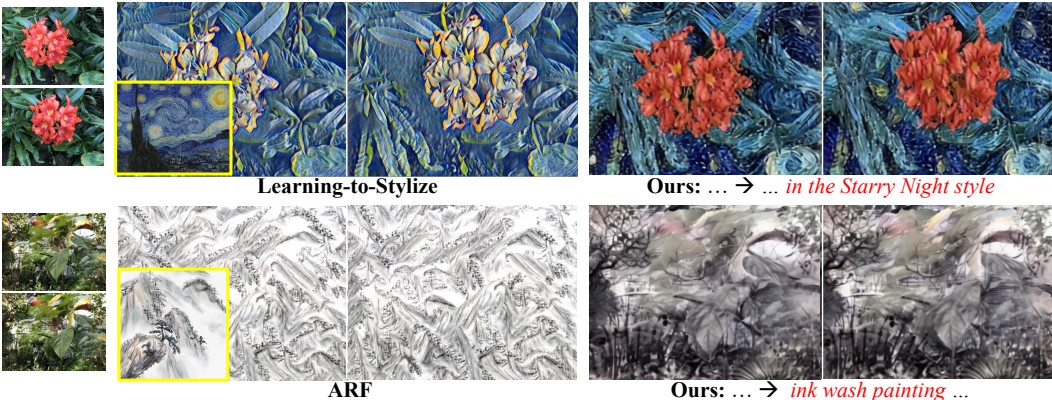

Figure 5: **Comparison with other methods for editing scene styles**. DN2N can more accurately transfer colors and brush strokes while preserving more original image content.

tion. More editing results are presented in the supplementary material. To further assess the effectiveness of our approach, we conduct comparative studies against state-of-the-art methods.

**Text-driven 3D scene editing**. We compare our approach to other text-driven editing methods, as shown in Figure 3. NeRF-Art utilizes multiple loss functions to retrain a model corresponding to given words, with limitations in achieving complex and precise editing effects. Instruct-N2N initially edits 2D images according to instruction and trains a model using these edited images, then generates new images by the model, which are further edited again to train the model iteratively until it produces images that comply with the instructions. In contrast, the scene editing results by our DN2N are directly inferred by the model after the given text without any intervening training process, whereas NeRF-Art and Instruct-N2N require retraining a model for each scene and each editing type. In addition, the editing results by DN2N are more realistic and retain more areas unrelated to the target caption, as shown in Figure 3, making it easier to achieve cascading multiple edits (see the supplementary material for more editing results).

**Appearance editing**. Two steps are involved in editing the appearance of an object in a scene: determining the target area and editing the appearance of that area. As illustrated in Figure 4, our method accurately locates the target area and applies appearance editing consistent with the target description without requiring training. While Clip-NeRF uses CLIP Similarity to limit the novel view to the target words, it cannot accurately locate the target area or transfer visual appearance. DFF uses DINO or Lseg to inject label information into points in space, ensuring precise editing area localization. However, its appearance editing requires manual operations, such as specifying the RGB value of the editing area. Furthermore, both methods require training separate models for each scene, making them less efficient than DN2N.

**Style transfer**. Figure 5 presents visual comparisons rendered by DN2N and other 3D scene style transfer techniques. As depicted, the Learning-to-Stylize method tends to imitate the color information of the reference image, but it often overlooks curved strokes. On the other hand, ARF excessively imitates curved strokes of the reference image, resulting in less pleasing visual effects and loss of information in the original scene. In contrast, our approach synthesizes the scenes by capturing both the color and stroke of the target style. Furthermore, both evaluated methods necessitate selecting a reference image first and then training an editing model, whereas our approach is text-driven and user-friendly, without retraining. As such, our method reduces training requirements and model storage, thereby offering friendly image editing capabilities.

## 4.2 QUANTITATIVE RESULTS

**Ability to resist perturbations**. Our method is specifically designed for scene editing, capable of maintaining 3D consistency under minor editing perturbations. To demonstrate this, we compare our approach to commonly used generalization models in 8 scenes on the LLFF dataset. Table 1 shows the results of two types of comparisons: one on unedited scenes (LLFF) and the other on scenes with minor editing perturbations (LLFF*) that result in some inconsistency. It can be seen that our

Table 1: **Comparison of mean PSNR with other generalizable NeRF models on the LLFF dataset.** * indicates random minor perturbations to the scenes.

|  | LLFF | LLFF* |
|---|---|---|
| PixelNeRF | 18.66 | 11.03 |
| MVSNeRF | 21.18 | 16.74 |
| IBRNet | 25.17 | 20.05 |
| Neuray | **25.35** | 19.31 |
| DN2N | 23.81 | **22.42** |

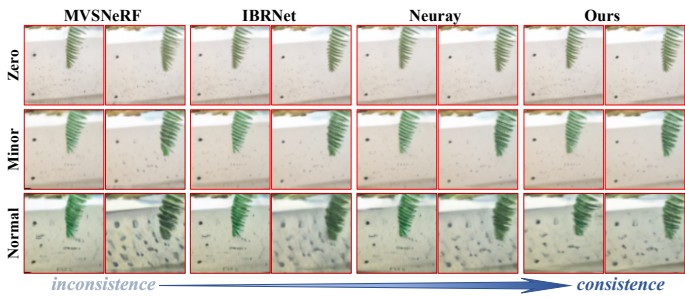

Figure 6: **Comparison with other generalizable NeRF models on the fern scene**. Our approach can maintain 3D consistency across novel views under different editing degrees.

Table 2: **Comparison of model capabilities, runtime, and model size for editing the flower scene**. 'TC' and 'SC' are Time and Space Complexity, respectively.

| Capabilities | | | | | Methods | Time (minute) | | | TC | Size(MB) | SC |
|---|---|---|---|---|---|---|---|---|---|---|---|
| generalizable | text driven | object editing | appearance editing | weather transfer | style transfer | | train | edit | total | | | |
|  |  |  |  |  | ✓ | **ARF** | 21.7 | **3.4** | 25.1 | O(n) | 558 | O(n) |
|  |  |  | ✓ |  |  | **DFF** | 20.6 | 5.2 | 25.8 | O(n) | 144 | O(n) |
|  | ✓ | ✓ | ✓ |  |  | **Clip-NeRF** | 524.4 | 349.7 | 874.1 | O(n) | 29.4 | O(n) |
|  | ✓ | ✓ |  |  | ✓ | **NeRF-Art** | 1545.6 | 780.6 | 2326.2 | O(n) | **18.3** | O(n) |
|  | ✓ | ✓ | ✓ | ✓ | ✓ | **Instruct-N2N** | 19.2 | 62.1 | 81.3 | O(n) | 484 | O(n) |
| ✓ | ✓ | ✓ | ✓ | ✓ | ✓ | **Ours** | **0** | 22.3 | **22.3** | O(n) | 103 | **O(1)** |

approach outperforms other models on scenes with minor inconsistencies resulting from 2D editing. Figure 6 also illustrates that our method can achieve superior 3D consistency in editing outcomes. These demonstrate our method is more suitable for the 3D scene editing task.

**Model efficiency**. Our generalizable model precludes retraining and has a variety of editing capabilities, which is more efficient and practical. A comparison of model capabilities and efficiency has been incorporated, and the results, as presented in Table 2, show substantial advantages of the proposed DN2N over previous techniques in capability, runtime, and model storage. In terms of runtime, our approach does not require training time for new scenes or new types of edits, only necessitating inference time for editing. Thus, our method is more efficient than alternative techniques. Regarding model storage, our model, which is applicable across all scenes and types of edits, entails a constant space complexity of $O(1)$. In contrast, other methods need to retrain a model for each different scene or type of edit, leading to space complexity of $O(n)$. Thus, these methods consume significantly more storage than our model when there are numerous editing scenes or editing types.

### 4.3 USER STUDY

Since scene editing is a subjective task, we perform a user study to provide a more generalized evaluation of the editing results of the DN2N method against state-of-the-art approaches. The study yields 1700 votes in total for three evaluation metrics: 3D consistency, preservation of the original scene content, and faithfulness to the text description. The results are depicted in Figure 7, which shows that the proposed method is favored in terms of these evaluation metrics. The implementation details of the user study can be found in the supplementary materials.

### 4.4 ABLATION STUDIES

We demonstrate the contribution of each component in our method in Figure 8. We find that omitting the data generation process during training would significantly affect the model performance on edited results. Furthermore, removing the self and neighboring view components leads to 3D inconsistency. Upon testing the content filter during network inference, we find that its absence results in significant image distortion. This can be attributed to the fact that editing 2D images directly results

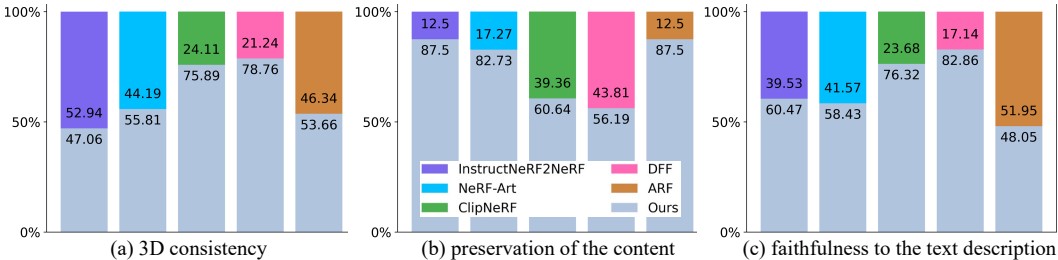

Figure 7: **User study**. The proposed DN2N method performs well against other state-of-the-art approaches in terms of comprehensive performance across three evaluation criteria.

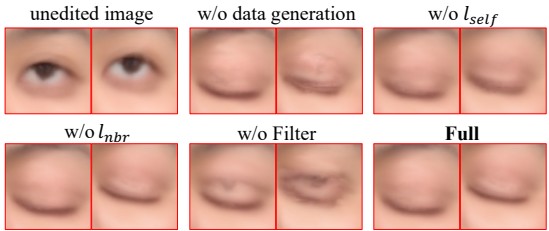

Figure 8: Qualitative ablation studies for key components in editing the 'open eyes' to 'closed eyes'.

Table 3: **PSNR results of ablation studies.** Experiments are conducted by applying varying minor perturbations to the scene in Figure 8. DG denotes data generation.

|      | w/o DG | w/o $l_{self}$ | w/o $l_{nbr}$ | Full      |
|------|--------|----------------|---------------|-----------|
| exp1 | 17.24  | 19.89          | 21.05         | **21.76** |
| exp2 | 18.98  | 21.5           | 21.92         | **23.11** |
| exp3 | 18.73  | 21.16          | 22.39         | **24.21** |
| exp4 | 18.97  | 20.53          | 22.19         | **22.27** |

in a significant 3D inconsistency between different views, making it difficult for the model to solve these inconsistencies without the content filter. Table 3 shows the quantitative ablation results by assessing the model resistance to disturbance by applying minor perturbations to the scene in Figure 8. It is evident that training the generalization model with perturbed images is crucial. Furthermore, enforcing cross-view consistency can also enhance the model's overall performance.

## 4.5 LIMITATIONS

Similar to the limitations encountered by the InstructN2N method (Haque et al., 2023), our outcome of the 3D editing is also affected by the 2D image pre-editing process. As illustrated in Figure 9, the 2D editing model may not always achieve reliable editing results, leading to failures in editing 3D scenes.

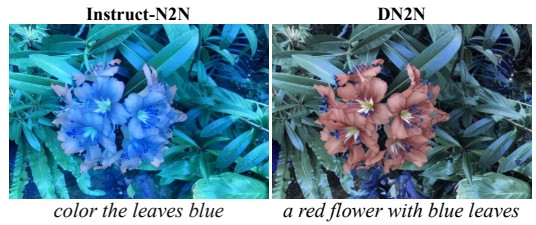

Figure 9: **Failed editing cases**.

## 5 CONCLUSION

In this work, we propose a text-driven method for editing 3D scenes that exhibits strong generalization capabilities and enables realistic novel view editing without additional training for each modification task. Our approach leverages existing 2D editing models to perform initial editing of 3D scene images based on textual descriptions. We then filter out poorly edited images that disrupt 3D consistency significantly, treating the remaining inconsistency as noise perturbation on top of consistently edited results. We provide several approaches to train our model to eliminate this perturbation, including creating training data and strengthening cross-view robustness. Our experimental results demonstrate the effectiveness of our approach, which offers significant advantages over conventional methods. Specifically, our method is more user-friendly for editing, supports multiple editing capabilities, and exhibits strong generalization. It eliminates the need for training on new scenes or editing types, resulting in a shorter editing time and lower memory overhead.

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

## A  BACKGROUND OF NEURAL RADIANCE FIELDS AND DIFFUSION MODELS.

**Neural radiance fields**. NeRF utilizes an implicit function to represent scenes, which maps the spatial point $\mathbf{x} = (x, y, z)$ and view direction $\mathbf{d} = (\theta, \phi)$ to the density $\sigma$ and color $\mathbf{c}$. Typically, the implicit function is represented by an MLP network, denoted as $F_\Theta : (\mathbf{x}, \mathbf{d}) \longrightarrow (\sigma, \mathbf{c})$, where $\Theta$ represents the weights of the network. For a ray $\mathbf{r}$ originating at $\mathbf{o}$ with direction $\mathbf{d}$, the RGB value $\hat{\mathbf{C}}(\mathbf{r})$ of the corresponding pixel is estimated through numerical quadrature of the color $\mathbf{c}_i$ and density $\sigma_i$ of the spatial points sampled along the ray:

$$\hat{\mathbf{C}}(\mathbf{r}) = \sum_i^N T_i (1 - \exp(-\sigma_i \delta_i)) \mathbf{c}_i, \tag{10}$$

where $N$ is the number of sample points, $\delta_i$ denotes the distance between adjacent sample points, and $T_i = \exp(-\sum_{j=1}^{j=i-1} \sigma_j \delta_j)$. Here $T_i (1 - \exp(-\sigma_i \delta_i))$ is used to calculate the entropy loss as defined in Eq. 7 in the main text.

**Text-guided diffusion models.** Text-guided diffusion models aim to generate an output image $z_0$ from a random noise vector $z_t$ under a textual condition $\mathcal{P}$. To achieve sequential denoising, the model $\epsilon_\theta$ is trained to predict artificial noise, minimizing the objective:

$$\min_\theta E_{z_0, \epsilon \sim N(0, I), t \sim \text{Uniform}(1, T)} \| \epsilon - \epsilon_\theta(z_t, t, \mathcal{C}) \|_2^2, \tag{11}$$

where $\mathcal{C} = \psi(\mathcal{P})$ denotes the embedding of the text condition, and $z_t$ is a noised sample according to the timestamp $t$. During the inference process, given a noise vector $z_T$, its noise is gradually removed by sequential prediction using a trained network for $T$ steps, which can be achieved by the DDIM sampling strategy (Song et al., 2021a). Amplifying the effect induced by the conditioned text is a significant challenge in a text-guided generation. To address this issue, Ho et al. (Ho & Salimans, 2022) propose a guidance technique that eliminates the need for a classifier in unconditional prediction and extends it to conditioned prediction scenarios. With the introduced concept of a null text embedding, denoted as $\varnothing = \psi("")$, and a guidance scale parameter $w$, then the guidance prediction is given by:

$$\tilde{\epsilon}_\theta(z_t, t, \mathcal{C}, \varnothing) = w \cdot \epsilon_\theta(z_t, t, \mathcal{C}) + (1 - w) \cdot \epsilon_\theta(z_t, t, \varnothing). \tag{12}$$

In our experiments, we primarily control the degree of image editing by adjusting the parameter guidance scale parameter $w$ and inversion steps $T$.

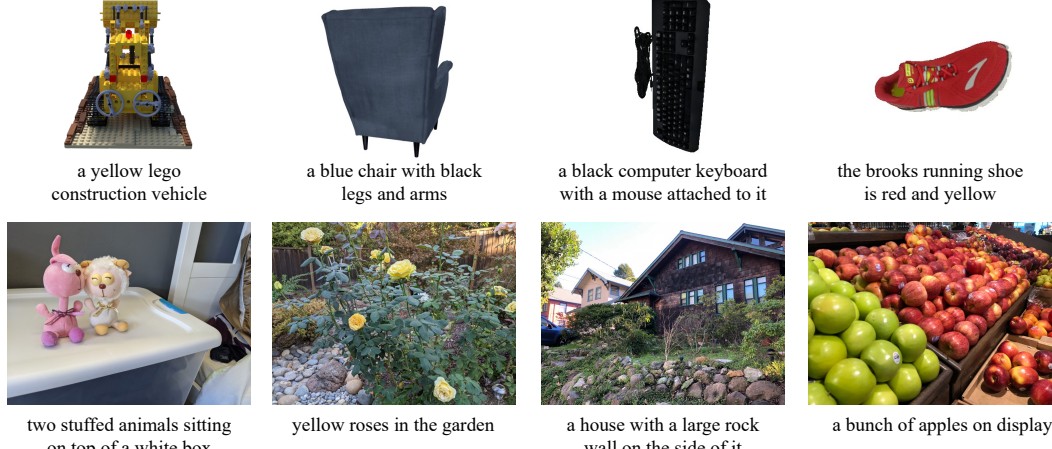

Figure 10: **Generate input captions by BLIP**. For each 3D scene, we randomly select one image to generate its input caption using the BLIP-2 model.

Table 4: **Generate target captions by GPT**. Given the input caption S for a 3D scene, we utilize GPT to generate a target caption O. For instance, if S is '*yellow roses in the garden* ', the target captions can be '*Leonardo da Vinci painting of yellow roses in the garden* ', '*yellow roses in the garden in the Rococo style*', '*pink roses in the garden* ', to name a few.

| Prompts | List 100 famous painters | List 50 famous painting schools | List 100 famous paintings | Replace, add or delete partial words in the following sentences: S1, S2, ... |
|---|---|---|---|---|
| GPT outputs (O) | Leonardo da Vinci
Vincent van Gogh
Pablo Picasso
Sam Francis
Max Ernst
.
.
Henri Matisse
Eva Hesse
Carl Andre
Cy Twombly
Jan van Eyck | Baroque
Realism
Impressionism
Op Art
Fauvism
.
.
Tonalism
Ashcan School
Rococo
Symbolism
Outsider Art | Mona Lisa
The Last Supper
The Scream
The Starry Night
Guernica
.
.
The Fifer
The Kiss
The Hay Wagon
Olympia
Sunflowers | pink roses in the garden
red roses in the garden
white roses in the garden
orange roses in the garden
purple roses in the garden
.
.
a green couch with gold trim
a green chair with silver trim
a green chair with no trim
a blue chair with gold trim
a white chair with gold paint |
| Target captions | O painting of S
or
S in the O style | O painting of S
or
S in the O style | S in the O style | O |

## B  ADDITIONAL IMPLEMENTATION DETAILS

### B.1  PROCESS OF GENERATING TRAINING DATA

**Generate input and target captions**. During the training phase, we utilized a total of 1246 scenes. For each scene, we randomly select one image to generate an input caption using the BLIP model (Li et al., 2023) with 2.7 billion parameters [1]. A subset of the generated captions is shown in Figure 10. Then we utilize the GPT model (Brown et al., 2020a) to generate target captions. As shown in Table 4, the four instruction prompts for GPT are:

(1). List 100 famous painters.

(2). List 50 famous painting schools.

(3). List 100 famous paintings.

(4). Replace, add, or delete partial words in the following sentences: X. (X is the input caption from BLIP.)

During training, we select one of the above (1)-(4) following a 2:2:2:4 ratio to generate the target caption. For (1), a chosen painter like "Van Gogh" could transform the caption "a red flower" into "Van Gogh painting of a red flower". Similar procedures apply to (2) and (3). For (4), GPT might change "a red flower" to "a red apple". Thus, each scene incorporates 405 target captions.

**Minor perturbations for training**. We generate the training data by applying minor random perturbations to the 3D scene using the 2D editing model Null-text [2]. The random ranges for the relevant parameters are set as follows: 100 to 300 for the iteration number $T$ and 0.5 to 3.5 for the text guidance scale $w$.

**Normal perturbations for inference**. For normal-scale edits, we follow the recommended settings from the Null-Text paper, with $w$=7.5 and $T$=500.

The effects of applying random minor and normal editing to the images are depicted in Figure 11.

---

[1] https://huggingface.co/Salesforce/blip2-opt-2.7b

[2] https://null-text-inversion.github.io

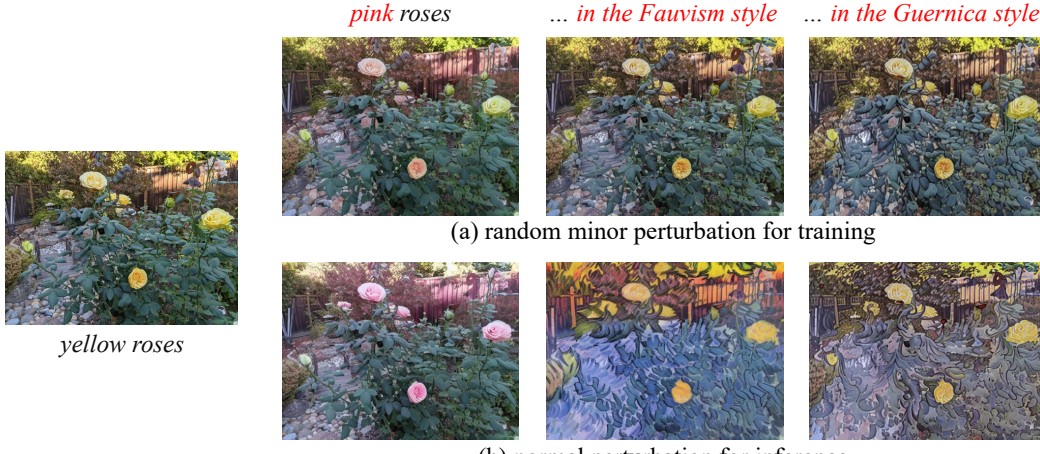

*pink roses*     *... in the Fauvism style*     *... in the Guernica style*

(a) random minor perturbation for training

*yellow roses*

(b) normal perturbation for inference

Figure 11: **Visualization of random minor perturbations and normal perturbations**. After applying 2D editing to the scene, the images exhibit various types of distortions in color or structure, thereby compromising the consistency among them. To simulate this 3D inconsistency, we introduce minor perturbations to the clean scene and optimize our model to remove them.

### B.2 Training and inference

**Training phase**. In the training process, the input caption for the current scene can be directly retrieved from a pre-stored file. As for the target caption, it is generated randomly based on Table 4. To be specific, we employ a weight of 2:2:2:4 to randomly choose one column from the four columns in Table 4 and then select its target caption for the current scene.

Our method is implemented using the PyTorch framework (Paszke et al., 2019). We employ the Adam optimizer (Kingma & Ba, 2014) with initial learning rates of 1e-4 for the CNN and 5e-4 for the MLP. The training process runs for 300K steps with a batch size of 500 rays. The initial values for the loss weight in Eq. 8 in the main text of $l_{self}$, $l_{nbr}$, $l_{en}$, and $l_{tv}$, are set as $\lambda_1$=1e-3, $\lambda_2$=1e-3, $\lambda_3$=1e-3, and $\lambda_4$=2e-3, respectively. The calculation of $l_{nbr}$ is only performed after the iteration count exceeds 10K. During the training phase, we randomly select a variable number of source views ranging from 6 to 15, while using 15 source views during inference. The number of sampled points on a ray is set to 64.

**Inference phase**. After completing the training process, given a scene and its corresponding textual description, we apply normal-level editing to the images of the scene using the 2D editing model. Following that, we employ a content filter to select the edited results, removing the lowest and highest 10% of values for each of the four metrics defined in Eq. 9 in the main text.

**Implementation details of the content filter**. The main idea behind the content filter is to maintain the degree of editing consistent across various perspectives. To achieve this, we consider the following two situations:

(1). A smaller degree of editing implies fewer changes in the edited image relative to the original, while a larger discrepancy with the target text.

(2). Conversely, a higher degree of editing denotes more significant changes in the edited image compared to the original, bringing it closer to the target text.

Therefore, the evaluation metrics used in the content filter can be measured through the relationship between the original image, edited image, original text description, and target text description. Specifically, given an original image $I_i$ and its caption $C_{in}$, as well as its corresponding edited image $\tilde{I}_i$ and its caption $C_{tgt}$, we calculate the following four measurements based on SSIM similarity and CLIP similarity during the filtering process:

(a). $\text{SSIM}(I_i, \tilde{I}_i)$

(b). $\text{CLIP}(I_i, \tilde{I}_i)$

(c). $\text{CLIP}(\tilde{I}_i, C_{tgt})$

(d). $\text{CLIP}(I_i, \tilde{I}_i) - \text{CLIP}(C_{in} - C_{tgt})$

Here, the metrics (a) and (b) assess the similarity between the image before and after editing. The metric (c) gauges the similarity between the edited image and the target caption, while the metric (d) evaluates the relative offset between the text and the image. These indicators assess the editing results on different dimensions. The role of our content filter is to eliminate extreme values (top 10% and bottom 10% of them) from these metrics, ensuring that the remaining images tend toward consistent editing results.

In Figure 12, we provide the maximum, minimum, and median values for each of the aforementioned metrics, along with their corresponding original and edited images. Figure 12 shows that the editing results with maximal and minimal metric values exhibit larger image discrepancies. These will pose greater challenges for subsequent 3D consistency. Our content filter, by eliminating these extreme editing results, facilitates superior 3D editing outcomes.

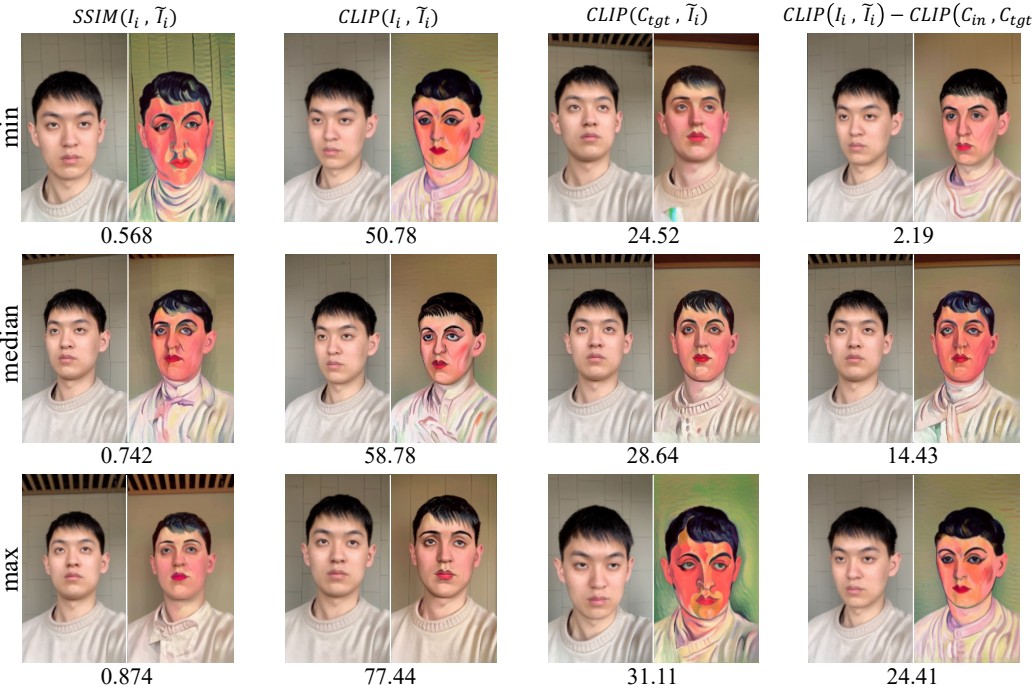

Figure 12: **Visualize the editing results of the 2D editing model, as well as the maximum, minimum, and median values for each metric of the content filter**. The images that correspond to the minimum and maximum values often exhibit either low or excessive editing degrees. By excluding them, we can enhance the 3D consistency among the remaining images.

## C  NETWORK ARCHITECTURE

Initially, we utilize a UNet-like network with the ResNet34 (He et al., 2016) backbone to extract features from both the unedited and edited source views. These feature maps are then concatenated and inputted into subsequent CNN networks to derive the final feature maps. When estimating the color **c** and density $\sigma$ of sampled points along the rays, we primarily employ the MLP networks to integrate, extract, and interpret information from the edited images, feature maps, and viewing directions. For density prediction, we adopt a design of transformer networks inspired by IBRNet (Wang et al., 2021). Regarding the color estimation, we incorporate additional pixel difference information among source views to capture the extent of editing across different perspectives, aiding the network in improved learning. The detailed network architecture and data propagation process are depicted in Figure 13.

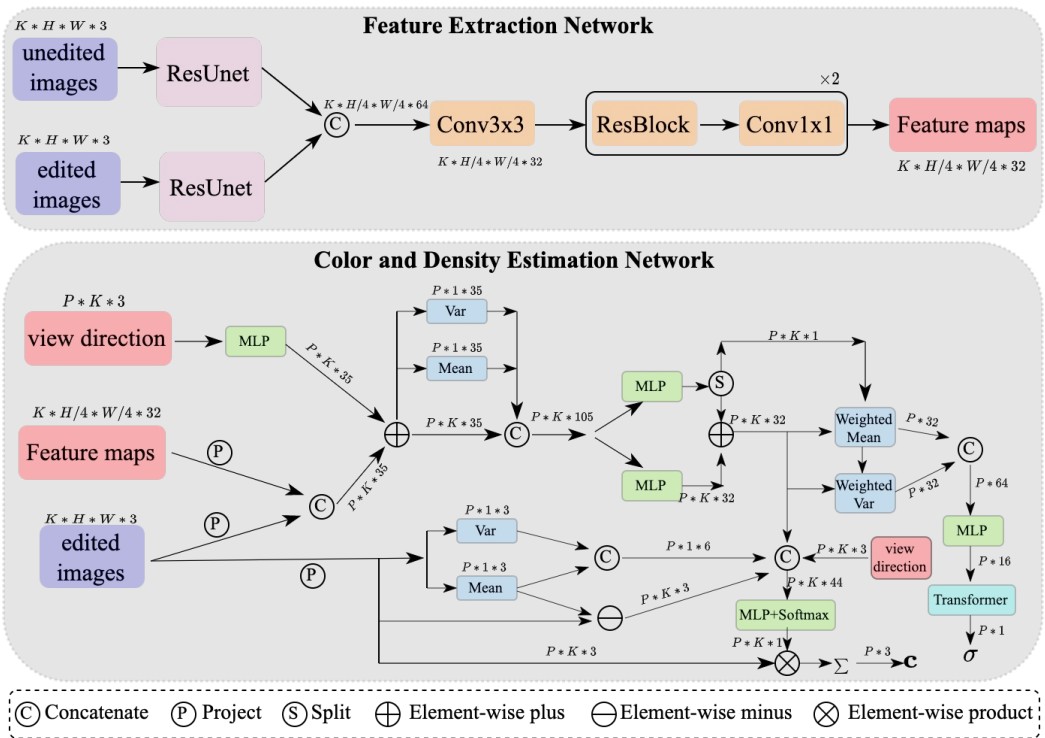

Figure 13: **Network architecture**. $P$ is the number of sample points on a ray, and $K$ is the number of source views.

Table 5: **Quantitative comparison of CLIP Directional Score (CDS) and CLIP Consistency Score (CCS)**. CDS* indicates using a different caption from CDS. CCS* means our result at the minimum editing degree. *The displayed values are multiplied by 100*.

|  | Van Gogh | | | | Fauvism | | | |
|---|---|---|---|---|---|---|---|---|
|  | CDS | CDS* | CCS | CCS* | CDS | CDS* | CCS | CCS* |
| NeRF-Art | **13.07** | 10.21 | 2.06 | 2.06 | **17.38** | 12.74 | 1.38 | 1.38 |
| Instruct-N2N | 11.21 | 10.39 | **9.24** | **9.24** | 16.62 | 13.18 | **4.11** | 4.11 |
| Ours | 8.83 | **11.76** | 4.89 | **100** | 14.06 | **14.93** | 2.29 | **100** |

## D    ADDITIONAL QUANTITATIVE COMPARISON

Although the evaluation of the editing results is subjective, we follow the descriptions and implementation codes of CLIP Directional score (CDS) and CLIP Consistency score (CCS) as outlined by Instruct-N2N (Haque et al., 2023) to quantitatively evaluate the editing results shown in Figure 3 of the main text. The quantitative results are presented in Table 5. It is clear that, under different settings, there are large variances in these metrics.

The CDS measures how much the change in text captions agrees with the change in images. When using CDS, a text description of the edited scene needs to be provided. Different descriptions yield varying results, as shown by CDS and CDS* in Table 5. This is mainly because the prompts used in training vary across different methods. For example, NeRF-Art just uses target words like "Van Gogh", and incorporates the CDS into the loss function (as in Equation 4 of the NeRF-Art paper), while Instruct-N2N employs instructional prompts such as "Make him look like Vincent Van Gogh". Our method, on the other hand, provides a target description like "Vincent Van Gogh is in front of the wall." Thus, providing an equitable text description for comparing CDS presents a challenge. With the description "Portrait of Van Gogh", NeRF-Art has higher CDS but worse visual results, as

presented in Figure 3. When "Vincent Van Gogh is in front of the wall" is provided, our method performs the best. Therefore, CDS may not fully evaluate image editing performance effectively.

The CCS measures the cosine similarity of the CLIP embeddings of each pair of adjacent frames in a novel camera path. This metric heavily depends on the degree of editing. In extreme cases where the edited result is identical to the unedited one, the CCS has a maximum value of 1. However, under such conditions, the editing effect is not achieved. Therefore, the CCS also has certain limitations.

Due to these considerations, although our method and Instruct-N2N have both visually compared with other text-driven editing methods, neither uses CDS and CCS for performance evaluation. Effective quantitative metrics for editing results remain challenging, so existing methods still opt for subjective evaluations, for instance, by conducting user studies to aggregate subjective evaluation results from multiple individuals in order to reflect the quality of the editing results.

## E  ADDITIONAL ABLATION STUDIES

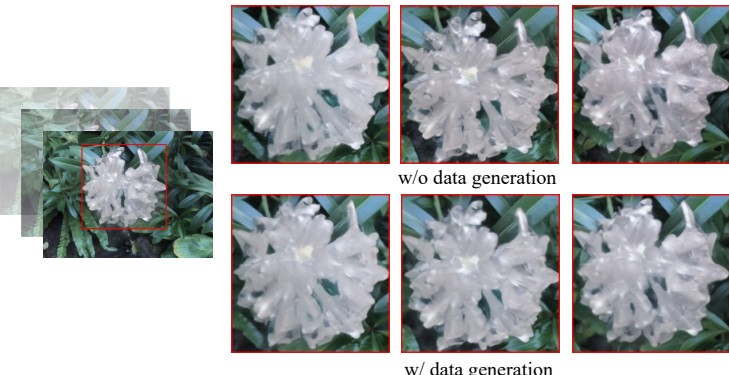

Figure 14: **Ablation study on data generation**. When training our generalization model, without incorporating data augmentation involving minor perturbations, the rendered novel views may exhibit noticeable inconsistencies in terms of glossiness and color.

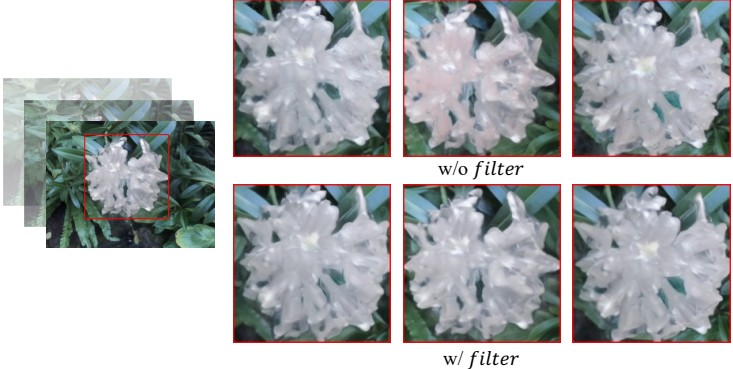

Figure 15: **Ablation study on content filter**. Removing the content filter component, the poorly edited images can detrimentally impact the final edited novel views, resulting in significant color inconsistencies, among other issues.

In this section, we present additional visualization and ablation experiments related to the training data generation, content filter, self-view ($l_{self}$), and neighboring-view ($l_{nbr}$) regularization terms. Specifically, we edit the flower scene in the LLFF dataset to a transparent ice sculpture flower.

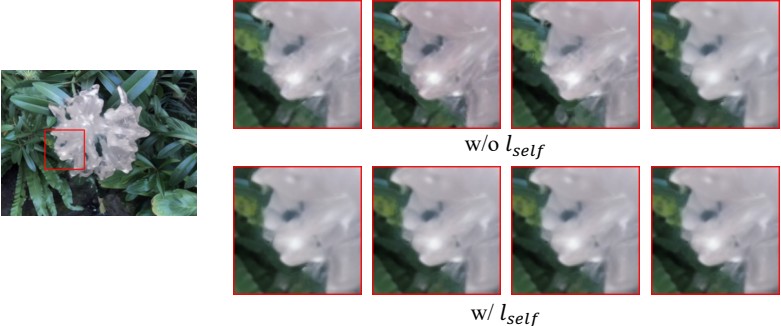

Figure 16: **Ablation study on $l_{self}$ regularization term**. The 3D consistency of the rendering results under different source views is poor (inconsistent color). While the introduction of the $l_{self}$ regularization term leads to a significant improvement in consistency.

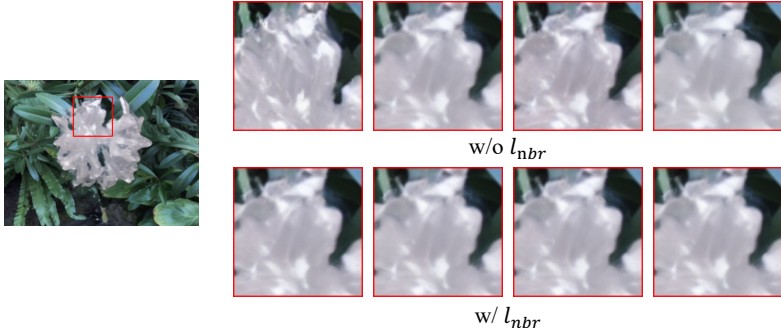

Figure 17: **Ablation study on $l_{nbr}$ regularization term**. A slight inconsistency (inconsistent gloss) exists among the results obtained from different neighboring views, which can be mitigated by introducing the $l_{nbr}$ regularization term.

During the training of our generalization model, we introduced inconsistent noise perturbations to the training data to encourage the model to learn the ability to remove inconsistencies. Upon removing this step, we observed significant color and gloss discontinuities between different novel views in the editing results as shown in Figure 14.

After training the model through data augmentation, it becomes capable of removing minor noise perturbations, while in the case of images with substantial consistency disruption in 2D editing results, this is mitigated by the content filter. Removing this filtering process leads to noticeable inconsistencies in the results as depicted in Figure 15.

We also visualize the results obtained by rendering with different source views and the rendering results of different neighboring views. The results are shown in Figure 16 and Figure 17. It can be observed that there is poor 3D consistency and varying degrees of color distortion in the results rendered under different source views. While after incorporating our designed $l_{self}$ regularization term, this phenomenon is significantly alleviated.

Similarly, there is a slight inconsistency among the results from different neighboring views, which is improved by introducing the $l_{nbr}$ regularization term.

## F    VISUALIZATION OF GEOMETRIC INFORMATION

We visualize the depth maps before and after scene editing, as shown in Figure 19. This is divided into two cases. First, when only appearance is edited, the depth maps before and after exhibit negli-

gible differences. Second, when the geometry of objects is altered, the depth maps correspondingly exhibit changes as well, validating the 3D awareness of our editing approach.

# G   ADDITIONAL EDITING RESULTS

In this section, we present a collection of additional editing results that showcase the remarkable editing capabilities and effects of our method. The results are shown as Figure 20, Figure 21, Figure 22 and Figure 23. Additionally, in Figure 24, we delve into more complex cascaded editing scenarios. These results demonstrate that our method can adapt to multiple types of datasets and has diverse editing capabilities.

# H   USER STUDY DETAILS

We invited a total of 50 subjects (31 males and 19 females, aged 18 to 45) to participate in the user study. Each participant is presented with videos or frames generated by various methods. They are asked to select their preferred option based on three distinct criteria: 3D consistency, content preservation, and faithfulness to the text description. The user study compared our method to others across 4 scenes with 18 questions. We use 23 videos to evaluate 3D consistency in 9 questions, and 24 images in the remaining 9 questions to measure the match between text prompts, editing results, and original image content preservation. Typical questions in the questionnaire are:
(1). What is the video with higher consistency?
(2). Please select an image that retains more of the original image content.
(3). Please select an image that more closely matches the text "Elf."

A screenshot of the questionnaire is available in Figure 18.

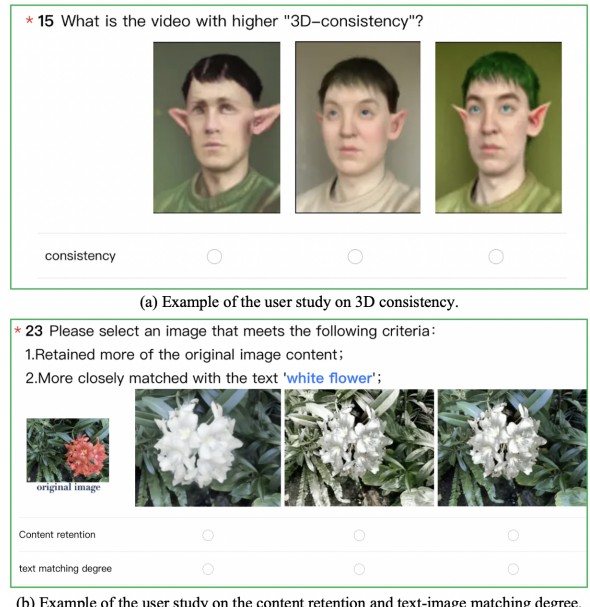

(a) Example of the user study on 3D consistency.

(b) Example of the user study on the content retention and text-image matching degree.

Figure 18: **Example of questionnaire for user study**.

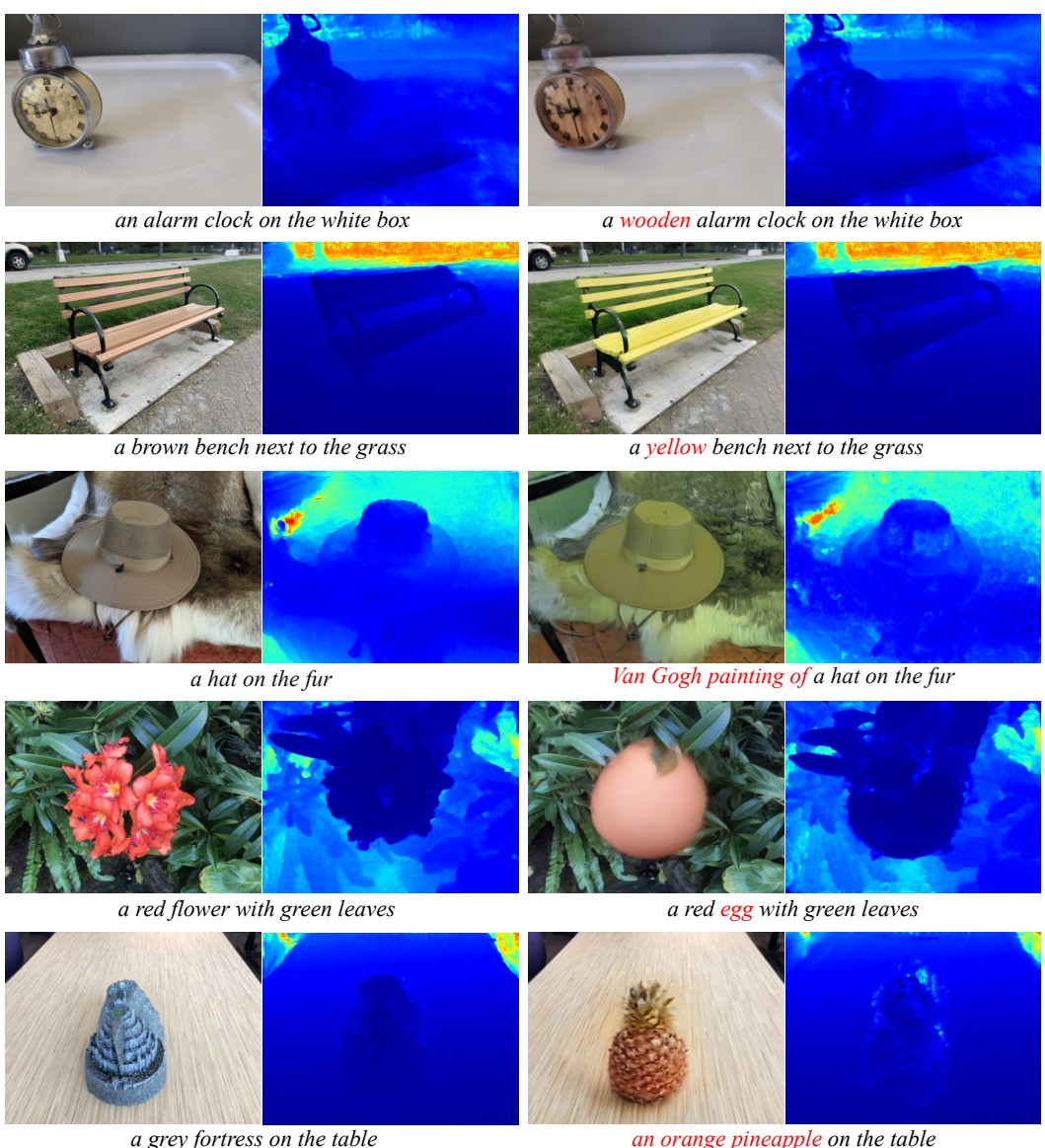

Figure 19: **Visualization of geometric information**.

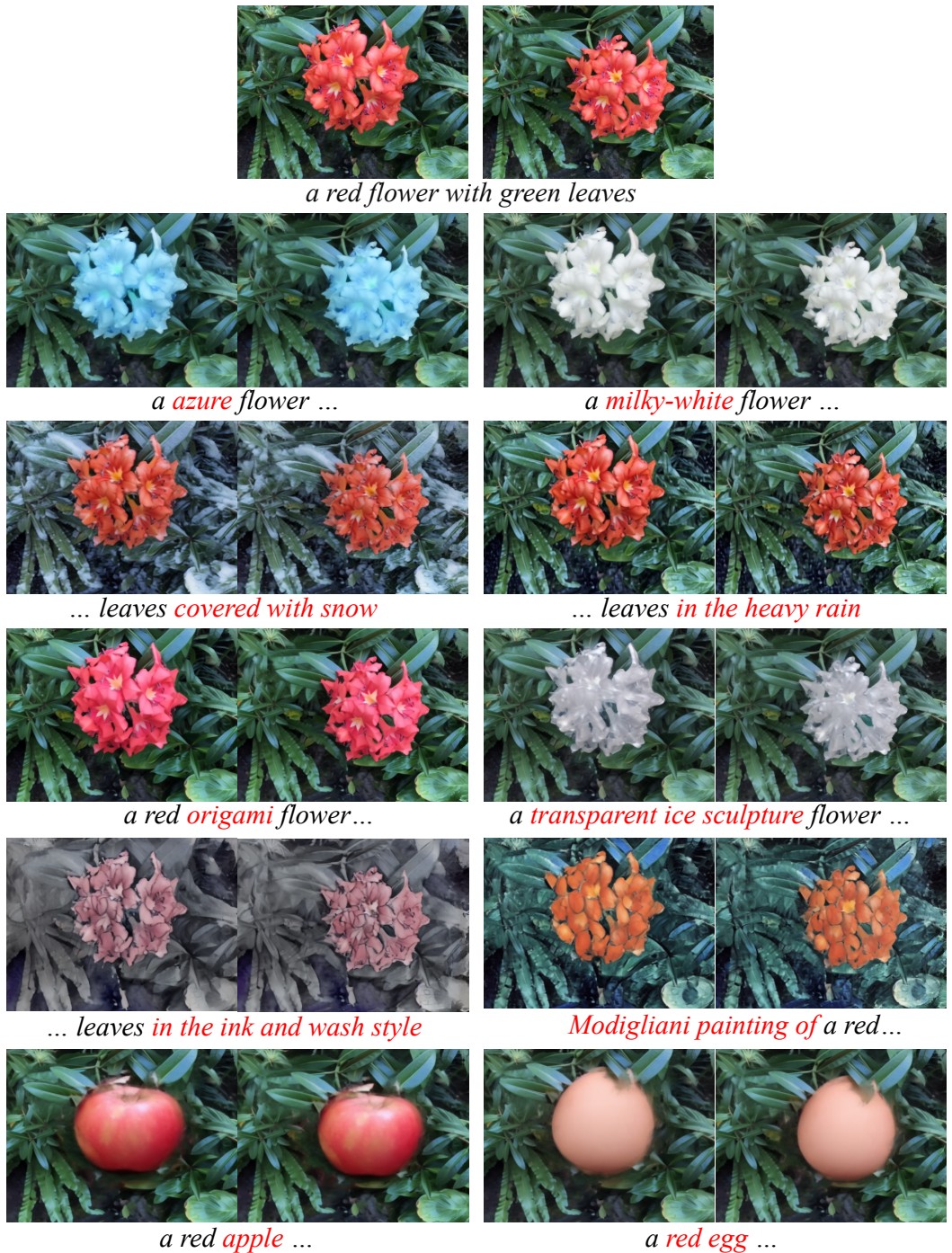

*a red flower with green leaves*

*a azure flower ...*          *a milky-white flower ...*

*... leaves covered with snow*          *... leaves in the heavy rain*

*a red origami flower...*          *a transparent ice sculpture flower ...*

*... leaves in the ink and wash style*          *Modigliani painting of a red...*

*a red apple ...*          *a red egg ...*

Figure 20: **More visual results on editing flower**.

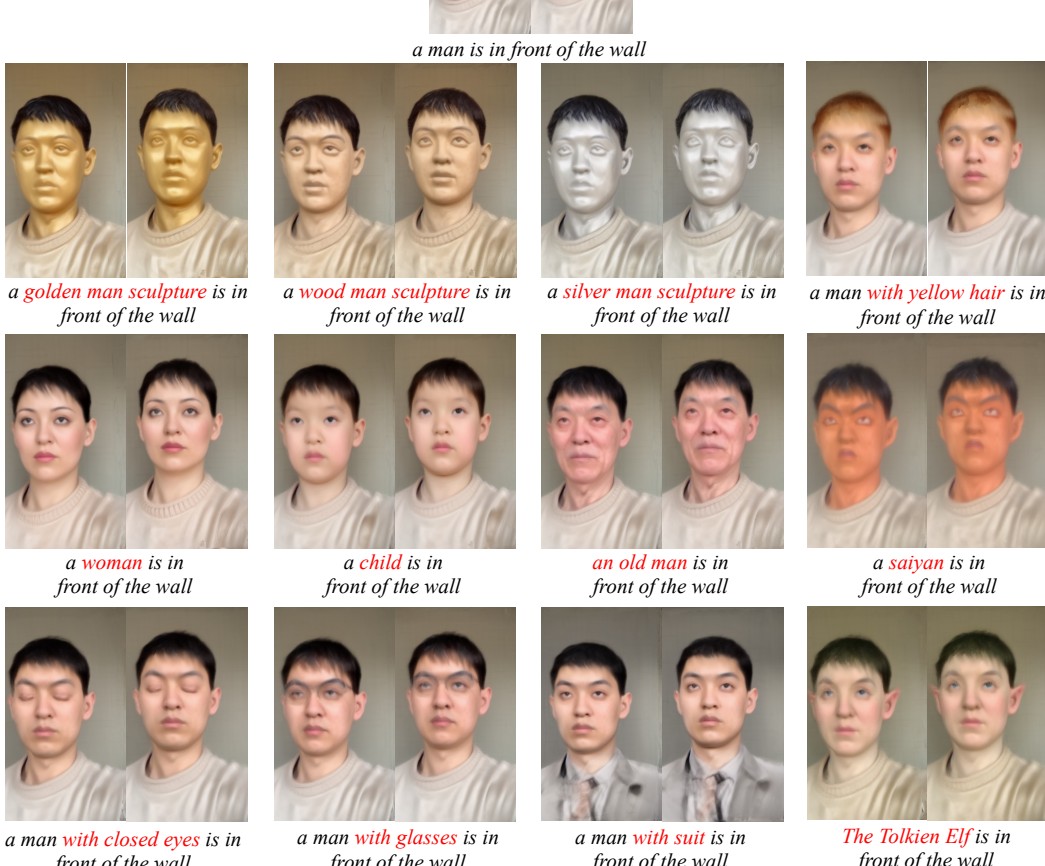

Figure 21: **More visual results on editing portrait**.

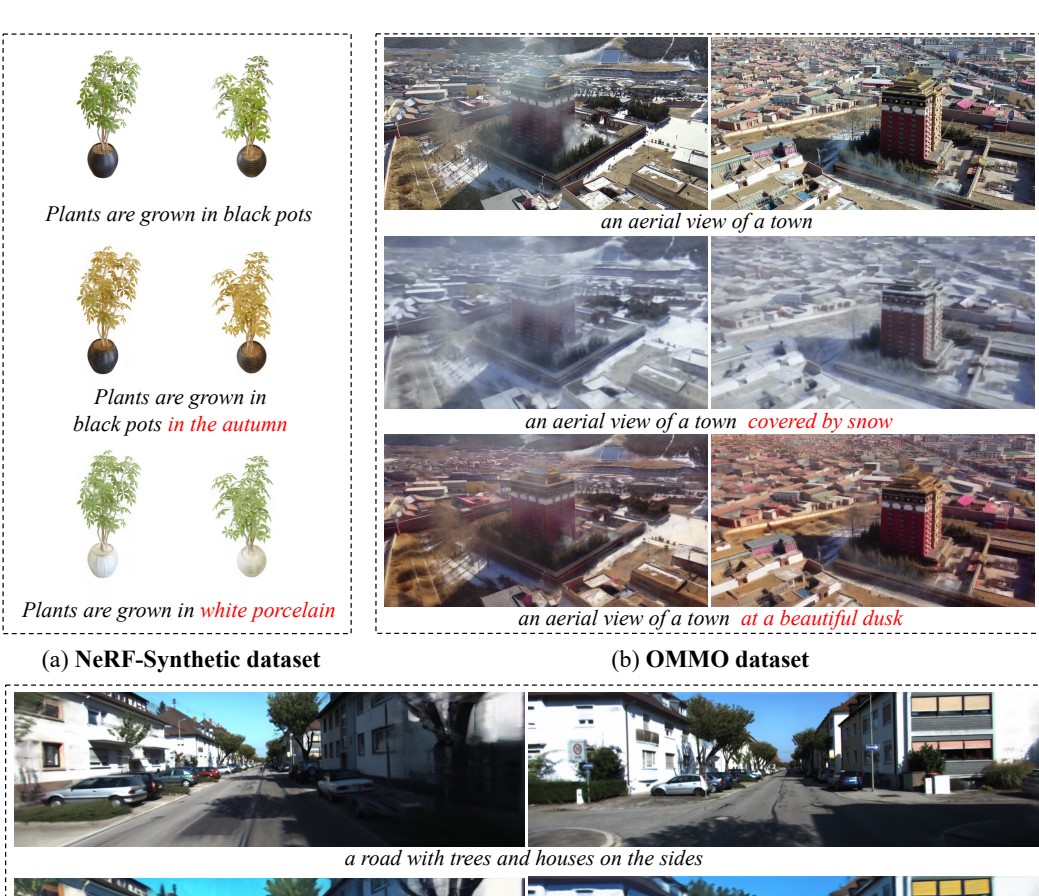

(a) **NeRF-Synthetic dataset**

(b) **OMMO dataset**

(c) **KITTI dataset**

Figure 22: **More visual results on NeRF-Synthetic, OMMO, and KITTI datasets**. The NeRF-Synthetic dataset (Mildenhall et al., 2020) is synthetic scenes. The OMMO dataset (Lu et al., 2023) is 360° scenes captured by the drone camera. The KITTI dataset (Geiger et al., 2013) is the street view scenes captured by the car camera.

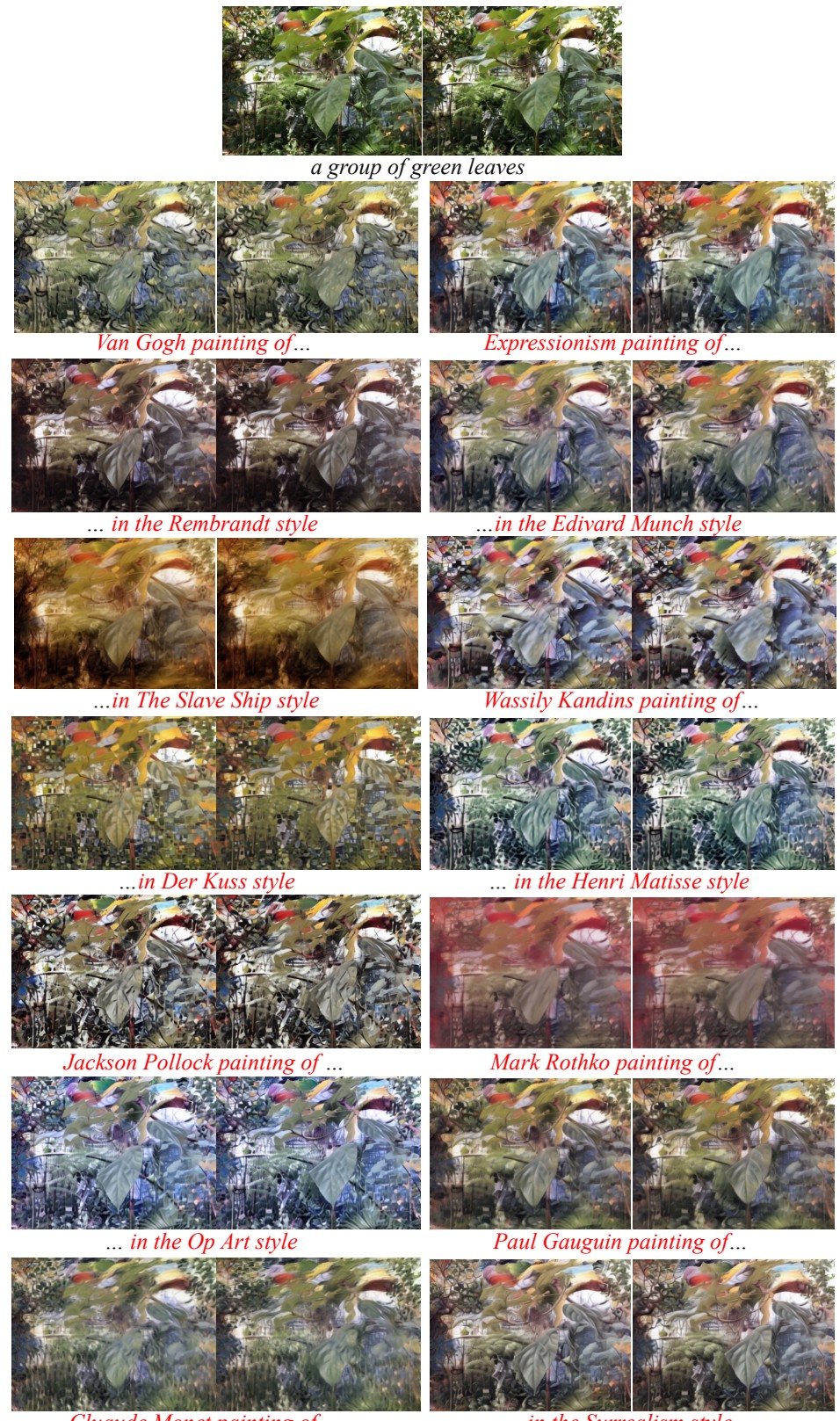

Figure 23: **More visual results on style transfer**.

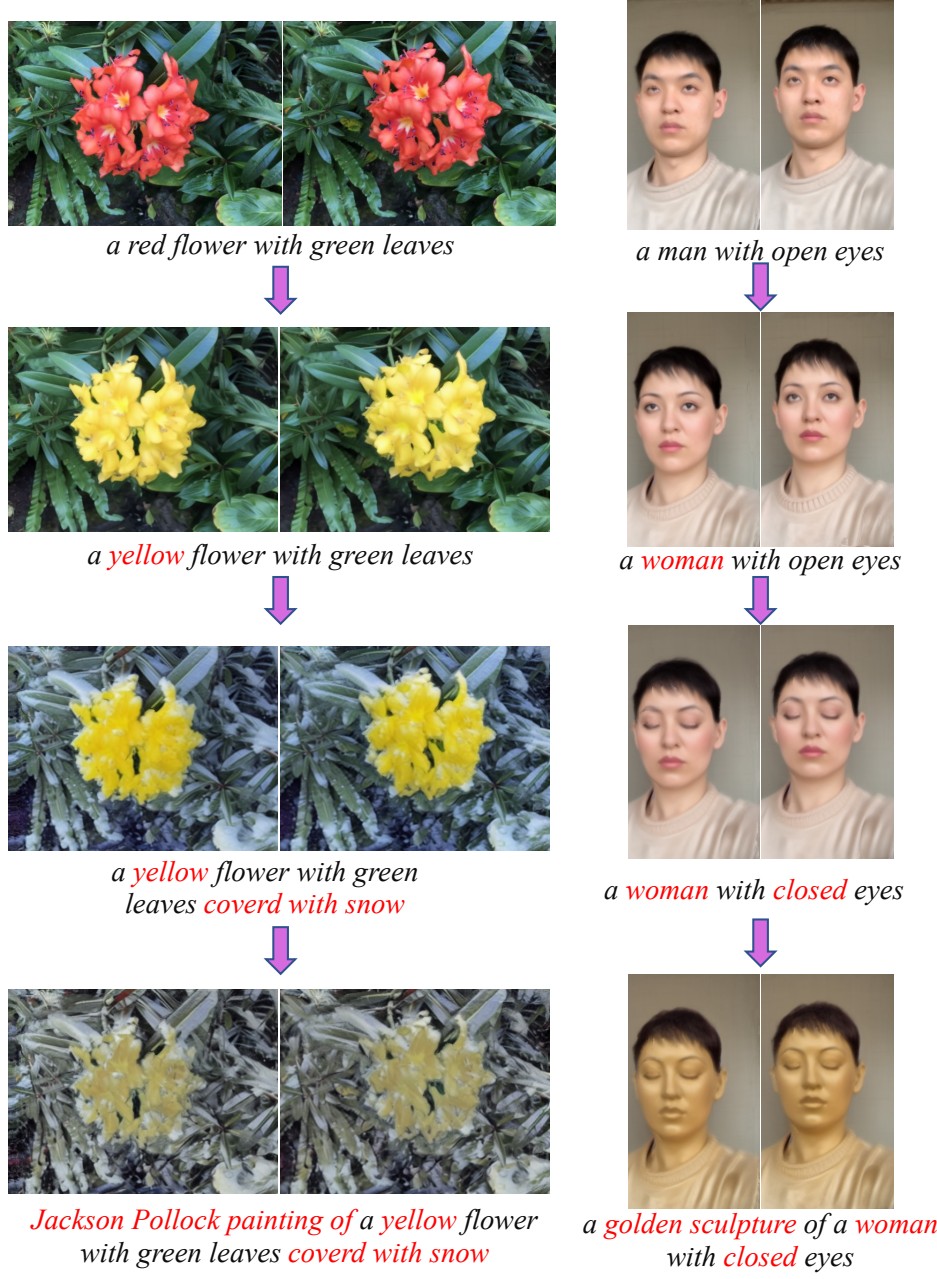

*a red flower with green leaves*

*a man with open eyes*

*a yellow flower with green leaves*

*a woman with open eyes*

*a yellow flower with green leaves coverd with snow*

*a woman with closed eyes*

*Jackson Pollock painting of a yellow flower with green leaves coverd with snow*

*a golden sculpture of a woman with closed eyes*

Figure 24: **Cascade editing results**. We begin by applying an edit to the original scene and generate a new scene with the desired editing effect using DN2N. We repeat this process by applying another edit to the new scene. In this figure, we showcase the consecutive results of three cascaded edits.

