# OpenReview forum: "Text-driven Editing of 3D Scenes without Retraining"
_ICLR.cc/2024/Conference — ICLR 2024 Conference Withdrawn Submission_

### Official Review · Reviewer_P4QJ · 2023-10-19

**Soundness:** 2 fair
**Presentation:** 2 fair
**Contribution:** 2 fair
**Rating:** 5
**Confidence:** 4

**Summary:**

To tackle the issues of specific design for various editing types, this work proposes framework that allows for the direct acquisition of a NeRF model with universal editing capabilities, eliminating the requirement for retraining. After modify the 3D scene images, a filtering process to
discard poorly edited images and use generalizable nerf to get consistent views. Cross-view regularization terms are used here.

**Strengths:**

extensive experiments of different editing types are conducted;

**Weaknesses:**

1. Generalizable nerf model can be limited to novel view generation in very limited view range;
2. The robustness of fliter technique is questionable to me
3. Abalation study cases are very limited.

**Questions:**

1. when using nearby view supervision, how to get depth is not well explained;
2. how to deduce from eq3 to eq4 is not clear to me
3. how many images used for generalizable nerf synthesis?

---

### Official Review · Reviewer_bafn · 2023-10-27

**Soundness:** 1 poor
**Presentation:** 3 good
**Contribution:** 3 good
**Rating:** 5
**Confidence:** 4

**Summary:**

The paper proposes a framework for text-driven 3D scene editing tailored with image-based rendering. The author claims to achieve generalizable editing capability which facilitates training-free editing. The key contribution is to generate lots of multiview training data by adding perturbations to 3D scenes using BLIP→GPT→Null-text inversion pipeline. Then, a generalizable image-based rendering model is trained to get rid of scene-specific training. The experiments show promising results on 3D editing. However, the generalization to novel text/scene is not validated.

**Strengths:**

- The idea to combine generalizable image-based rendering with 3D scene editing is interesting.
- The pipline BLIP→GPT→Null-text inversion to generate edited images on caption-free multiview dataset is promising.
- The visual result is compelling with good demonstration on various types of scenes, including object-centric scene, face-forward scene, and unbounded scene.

**Weaknesses:**

Given the lack of validation on the generalizability, I lean to rejection at this point. I’m happy to be convinced by the authors’ response.

- My major concern is the generalizability. According to the paper and supplementary material, I do not find any descriptions on the train/test split for the experiments. If all the results presented in the paper are seen during training, the paper definitely overclaims the generalizability. Thus, the title is also misleading as “without retraining”.
- I hope to see more details about the setting of experiments and corresponding evaluation protocols:
    - The train/test split of text captions. If all captions used in the qualitative results are seen during training, please present the result of editing the scene with unseen captions
    - The train/test split of 3D scenes. If all scenes presented in the qualitative results are seen during training, please exclude some of them and retrain the model. Then test on those unseen scenes. Otherwise, the method still needs retraining given user-specific 3D scenes, which does not support the main claim.

**Questions:**

- Please see the weakness section for my concern about generalizability.
- What is the motivation of src_a and src_b. Please clarify the motivation for these two separate models.

---

### Official Review · Reviewer_xjTT · 2023-11-01

**Soundness:** 1 poor
**Presentation:** 1 poor
**Contribution:** 2 fair
**Rating:** 1
**Confidence:** 3

**Summary:**

This paper uses diffusion model for 2D edits by text and consolidates the edits in NeRF. It claimed to work across several appearance editing and style transfer and also reduce the time overhead.

**Strengths:**

1. The paper contributes to quite a relevant NeRF editing topic with broad interest.
2. Extensive results are presented.

**Weaknesses:**

1.  As Introduction mentioned, existing methods typically rely on known editing types in advance with limited modification capabilities. Please elaborate on this claim and raise what editing types or some works who heavily rely on editing types. As of my knowledge and experience, Instruct-Nerf2Nerf or Clip-based methods are capable of any text input. They are not confined in specific editing types demonstrated in their paper.

"These techniques are often less user-friendly" seems semantically overlap with prior two claimed challenges. It does not add new information. Suggest to take off.

2. 2D editing does not guarantee multi-view consistency for each input. However, each edited single frame is well-structured and jumpy between multiview may not be simply noisy perturbations. For example, changing the selfie into Fauvism can change the selfie into multiple (colorful/ high contrast) possible output and each output is well-structured but quite different. The difference between diffusion model outputs may not be noisy perturbation.

3. The writing is complex and need much more clarity. For example, what is the purpose of input caption? Does it aims to provide a description and replace nouns or subjects to create a target caption for editing? What is the generalizable mean in the context?

4. There is not enough information for the inference stage. It only mentioned content filter in the paragraph. In Fig. 2 inference time, how can NeRF generate closed eye image if it has not seen examples in the training time? From the figure, after the volume rendering it shows an open-eye image, but the next step in G's output, it abruptly show closed eye image.

How to use the filtered image at the inference time to avoid re-training NeRF is also not clear to me.

5. The abstract claims doing appearance editing, weather transition, object changing, and style transfer. However, it seems the results only serve style transfer and appearance editing. It is not convincing to say changing to snow-covered roads is weather changes. The pineapple and strawberry examples seem just to change appearance only. It does not create new geometry.

**Questions:**

1. What exactly the "generalizable" term mean in NeRF? This term appears many times in the paragraph but is still vague. For example, what this NeRF model can attain that a vanilla NeRF model cannot.

2. In Eq. 6, how to get M for overlapping areas? Is it provided in the data?

3. In Fig.7, it mentioned a total of 6 methods for comparison but why in the exemplar Fig. 18 has only a total of 3 videos in comparison? (or is it just an example). How does one compute the ratios of one against the others in Fig. 7? It's hard to understand how the subjective test result in Fig. 7 is calculated. Besides, using only 4 scenes for a subjective test is far insufficient in my opinion. Also, please add a statistical significance p-value to validate the significance of the study.

---

> ### Author Response · Authors · 2023-11-13
> **Responses to questions 1 to 7**
>
> **Q1:** Please elaborate on this claim and raise what editing types or some works who heavily rely on editing types. As of my knowledge and experience,
> Instruct-Nerf2Nerf or Clip-based methods are capable of any text input and user-friendly.
>
> **R1:** In the introduction, we listed three potential flaws in existing methods. We want to clarify that each method has one or more of these flaws, rather than all having all three flaws. To summarize the types of flaws in each approach:
>
>   1. Rely on known editing types in advance: DFF, Learning-to-Stylize, ARF.
>   2. Require retraining an editing model for each specific 3D scene: CLIP-NeRF, Instruct-NeRF2NeRF, NeRF-Art, DFF, Learning-to-Stylize, ARF.
>   3. Less user-friendly: DFF, Learning-to-Stylize, ARF.
>
> Our main contribution in this paper is proposing a new editing framework that overcomes all these three flaws.
>
> **Q2:** Edited single frame is well-structured and jumpy between multiviews may not be simply noisy perturbations.
>
> **R2:** The differences between the outputs of diffusion models may not be noise perturbations, but they may have their own inherent patterns of these differences. The reason we refer to them as 'noise' perturbations in the article is simply to borrow the idea of 'denoising' for convenience: that is, denoising methods are used to remove noise, while our method is used to remove differences between the outputs of diffusion models.
> Regardless of what this difference specifically is, the purpose of training our model is to eliminate it, and our experimental results support the validity of this thought.
>
> **Q3:** The writing is complex and needs much more clarity. For example, what is the purpose of input caption?
>
> **R3:** The role of input and output caption is explained in Eq.1 of the article. Our method uses a 2D editing model called Null-text inversion. This method requires the provision of both input and target captions when editing images. The input caption represents the original image description, and the target caption denotes the desired image description.
>
> **Q4:** What is the generalizable mean in the context?
>
> **R4:** The 'generalizable' refers to the ability of our model to edit new scenes without the need for retraining.
> In contrast, methods such as Clip-NeRF and Instruct-NeRF2NeRF require retraining the model when editing new scenes.
>
> **Q5:** There is not enough information for the inference stage. It only mentioned content filter in the paragraph.
> In Fig. 2 inference time, how can NeRF generate closed eye image if it has not seen examples in the training time?
> From the figure, after the volume rendering it shows an open-eye image, but the next step in G's output, it abruptly show closed eye image.
> How to use the filtered image at the inference time to avoid re-training NeRF is also not clear to me.
>
> **R5**: Detailed explanations and examples of the inference and filtering process can be found on pages 16 and 17.
>
> Regarding the explanation of the transition from open eyes to closed eyes in Fig. 2:
>
> During the training phase (Fig.2a), we do not use closed-eye data as ground truth (GT) to train the model. The data pairs used for training are {the slightly perturbed image $I_{in}$ and the clean original image $I_{gt}$}. The goal of training our model is to remove the differences between these two images. Since only slight perturbations were applied during editing, the eye is not actually closed. Furthermore, this is just one type of edit on one scene. In our training stage, we used 1198 scenes, each constructing 405 training data pairs. Therefore, there are a total of 485,190 training data pairs for our model's training. After completing the training, our model has the ability to eliminate the differences between the output results of the 2D editing model.
>
> During the inference phase (Fig. 2b), we apply normal amplitude edits to images from 3D scenes. At this point, we first filter out some images with poor editing results, then use the model obtained in the previous step to remove the remaining perturbations. Thus, the inference phase can produce the result of closing the eyes.
>
> **Q6:** It is not convincing to say changing to snow-covered roads is weather changes.
>
> **R6:** Our method also simulates the effect of raindrops. Please see Supplementary Material Fig. 20, where the target caption reads "a red flower with green leaves in the heavy rain".
> We also referenced other articles such as "ClimateNeRF: Extreme Weather Synthesis in Neural Radiance Fields", presented at ICCV this year. Similarly, that article only covers snow weather effects.
>
> We will clarify this point in our final article version.
>
> **Q7:** In Fig.1, the pineapple and strawberry examples seem just to change appearance only. It does not create new geometry.
>
> **R7:** Geometric changes may not be obvious in this example, we suggest referring to Supplementary Material Fig. 20 in the article, where we demonstrate edits such as turning flowers into apples and eggs.

---

> ### Author Response · Authors · 2023-11-13
> **Responses to questions 8 to 10**
>
> **Q8**: What exactly the "generalizable" term mean in NeRF? What this NeRF model can attain that a vanilla NeRF model cannot.
>
> **R8**: Our model possesses the ability to generalize and render novel views for new scenes without retraining, whereas vanilla NeRF must be trained for each individual scene.
>
> For more information on the design of generalizable NeRF models, please refer to IBRNet, MVSNet, Neuray, and GeoNeRF.
>
> **Q9:** In Eq. 6, how to get M for overlapping areas? Is it provided in the data?
>
> **R9:**  This is achieved by projecting points from one viewpoint onto another viewpoint using depth information. For a certain region on viewpoint A, it may be invisible on viewpoint B, and thus will not be used in calculating the loss.
>
>
> **Q10:** Why in the exemplar Fig. 18 has only a total of 3 videos in comparison? (or is it just an example).
> using only 4 scenes for a subjective test is far insufficient in my opinion.
> How does one compute the ratios of one against the others in Fig. 7? Besides, please add a p-value to validate the significance of the study.
>
> **R10:**
> Figure 18 is just an example.
>
> Although we used 4 scenes, multiple types of edits were performed on each scene and we collected a total of 1700 votes. As a comparison, StylizedNeRF collected 1000 responses across 4 scenes, while ARF collected 1150 votes across 5 scenes.
>
> Regarding how to compute the ratios in Fig.7, here is a simple example:
> We showed users two videos (one generated by DN2N and one generated by CLIP-NeRF) and asked them to choose which video they thought had better consistency. If 30 users chose DN2N and 20 users chose CLIP-NeRF, the ratio of DN2N would be calculated as 30/(30+20) = 60% and the ratio of CLIP-NeRF would be calculated as 20/(30+20)=40%.
>
> We performed a binomial test on each category of voting results, and the p-values are listed below. In approximately half of the cases, there was a clear preference among users (with p-value < 0.05, indicating a tendency to prefer our model over others).
>
> |                      | (a).3D consistency | (b).preservation of the content | (c).faithfulness to the text description |
> |----------------------|----------------|-----------------------------|--------------------------------------|
> | p(DN2N, InstructN2N) | 0.887          | **3e-8**                        | 0.202                                |
> | p(DN2N, NeRF-Art)    | 0.479          | **5e-6**                       | 0.322                                |
> | p(DN2N, ClipNeRF)    | **0.0003**         | 0.202                       | **0.0003**                          |
> | p(DN2N, DFF)         | **9e-5**           | 0.479                       | **5e-6**                                |
> | p(DN2N, ARF)         | 0.671          | **3e-8**                        | 0.887                                |
>
>
>
> **Thank you for your reviews and comments. Please let us know if you have further questions after reading our response.**

---

### Official Review · Reviewer_A4Ej · 2023-11-02

**Soundness:** 2 fair
**Presentation:** 2 fair
**Contribution:** 3 good
**Rating:** 5
**Confidence:** 2

**Summary:**

This paper presents a way to perform editing of 3D scenes represented using NeRFs using text description of the target. They use BLIP for image captioning, GPT to generate target captions, image editing using text, etc., in the training step and train a generalisable NeRF model. During inference, text-based image editing is performed and the images that are 3D inconsistent are removed using a filtering step. The remaining images are used to generate new views using a network derived from IBRNet. The paper presents different editing results. The paper presents many ideas which are mostly explained in the long supplementary/appendix section.

**Strengths:**

Text-based editing of 3D scenes is an important problem and provides opportunities to create many variations of 3D representations if it can be done well. The paper leverages many recent methods to achieve this task in some way. They train the NeRF model using many more losses to achieve "generalisability". There are many good results shown.

**Weaknesses:**

Presentation/Writing: The presentation needs to improve as it is not clear what their training achieves. It is impossible to understand the method to a reasonable extent without reading the appendix or supplementary that runs into 15 pages. See my questions below.

Significance: While text-based editing will be useful, what is the predictability and controllability of such a method? How does one know if the generated model is according to the textual targets given? It seems there are few ways to do that beyond basic qualitative or visual assessments. This is a serious limitation of many methods that rely on a generative tool as there is little controllability or predictability on what such a tool generates. The DN2N method presented also has that problem also: what does the text-based-editing block generate at inference time? The method can only filter out inconsistent images. That is, it can only discard the images generated; it can't generate a better image by influencing the editing module. How do we know if selected good results only are shown here? The discussion in Section 4.5 needs to be far more elaborate.

**Questions:**

I have several doubts/questions about the inference or editing stage. These should be in the main paper clearly.
- What is the "effort" involved during inference? How many 2D edited images are generated? How many are found to be inconsistent on an average and discarded?  What is the time taken?
- Were there failure cases when sufficient # of consistent images couldn’t be generated? Is the whole process iterated on if that happens?
- Are edited images generated for the same camera poses of the input images used for training? Can other viewpoints be used?
- Is there any way to ensure or control what we want of the text-based image editing module? Your method is very critically dependent on this module generating good ones. Your editing capability is limited seriously by this aspect. Please see my comments under "weakness".
- The filtering step is rather too simple. How are the 4 measures used in the tuple prioritised or weighted? The details of the sorting could not be found anywhere. Why do you eliminate the top 10% of the matches? Aren't they the best?

Also, why is DN2N's output falling short in some methods in the user study?

**Details Of Ethics Concerns:**

None special beyond other methods that generate 3D objects which can be misused.